

# Harmonized gap-filled datasets from 20 urban flux tower sites

Mathew Lipson[1,2], Sue Grimmond[2], Martin Best[3], Winston Chow[4], Andreas Christen[5], Nektarios Chrysoulakis[6], Andrew Coutts[7], Ben Crawford[8], Stevan Earl[9], Jonathan Evans[10], Krzysztof Fortuniak[11], Bert G. Heusinkveld[12], Je-Woo Hong[13], Jinkyu Hong[14], Leena Järvi[15], Sungsoo Jo[16], Yeon-Hee Kim[17], Simone Kotthaus[18], Keunmin Lee[19], Valéry Masson[20], Joseph P. McFadden[21], Oliver Michels[22], Wlodzimierz Pawlak[23], Matthias Roth[24], Hirofumi Sugawara[25], Nigel Tapper[26], Erik Velasco[27], Helen Claire Ward[28]

[1]Australian Research Council (ARC) Centre of Excellence for Climate System Science, Climate Change Research Centre, Level 4, Mathews Building, UNSW Sydney, New South Wales, 2052, Australia.
[2]Department of Meteorology, University of Reading, Reading, RG6 6ET, United Kingdom
[3]Met Office, Fitzroy Road, Exeter, Devon, EX1 3PB, United Kingdom.
[4]School of Social Sciences, Singapore Management University, Singapore.
[5]Environmental Meteorology, Institute of Earth and Environmental Sciences, Faculty of Environment and Natural Resources, University of Freiburg, Freiburg, Germany.
[6]Foundation for Research and Technology Hellas, Institute of Applied and Computational Mathematics, Remote Sensing Lab, Heraklion, Greece.
[7]School of Earth, Atmosphere and Environment, Monash University, Melbourne, Australia.
[8]Geography and Environmental Sciences, University of Colorado, Denver, Colorado, USA.
[9]Global Institute of Sustainability and Innovation, Arizona State University, Tempe, Arizona, USA.
[10]UK Centre for Ecology & Hydrology, Crowmarsh Gifford, Wallingford, United Kingdom.
[11]Department of Meteorology and Climatology, University of Lodz, Lodz, Poland.
[12]Department of Meteorology and Air Quality, Wageningen University, The Netherlands.
[13]Korea Environment Institute, Sejong, Republic of Korea.
[14]Ecosystem-Atmosphere Process Lab, Department of Atmospheric Sciences, Yonsei University, Seoul, Korea (Republic of).
[15]Institute for Atmospheric and Earth System Research/Physics, Faculty of Science, University of Helsinki, Helsinki, Finland.
[16]Ecosystem-Atmosphere Process Lab, Department of Atmospheric Sciences, Yonsei University, Seoul, Korea (Republic of).
[17]National Institute of Meteorological Sciences, Korea Meteorological Administration, Jeju, Korea (Republic of).
[18]Institut Pierre Simon Laplace (IPSL), CNRS, École Polytechnique, Institut Polytechnique de Paris, 91128 Palaiseau Cedex, France.
[19]Ecosystem-Atmosphere Process Lab, Department of Atmospheric Sciences, Yonsei University, Seoul, Republic of Korea.
[20]Centre National de Recherches Météorologiques, University of Toulouse, Météo-France and CNRS, Toulouse, France.
[21]Department of Geography and Earth Research Institute, University of California, Santa Barbara, United States.
[22]Environmental Meteorology, Albert-Ludwigs-University, Freiburg, Germany.
[23]Department of Meteorology and Climatology, University of Lodz, Lodz, Poland.
[24]Department of Geography, National University of Singapore, Singapore.
[25]National Defense Academy of Japan.
[26]School of Earth, Atmosphere and Environment, Monash University, Melbourne, Australia.
[27]Independent Research Scientist, Singapore.
[28]Department of Atmospheric and Cryospheric Sciences, University of Innsbruck, Innsbruck, Austria.

*Correspondence to*: Mathew J. Lipson (mathew.lipson@unsw.edu.au)





**Abstract.** Twenty urban neighbourhood-scale eddy covariance flux tower datasets have been harmonized and quality controlled, producing a 50 site-year collection with broad diversity in climate and urban surface characteristics. Observations are gap-filled and prepended with 10 years of reanalysis-derived local data to enable use as spin up and forcing for land surface model evaluation. For both gap filling and spin-up, ERA5 reanalysis meteorological data are bias corrected using tower observations, accounting for diurnal, seasonal and local urban effects not modelled in ERA5. The bias correction methods developed perform well compared to methods used in other datasets (e.g. WFDE5 or FLUXNET2015). Site description metadata includes local land cover fractions (buildings, roads, trees, grass etc.), building height and morphology, aerodynamic roughness estimates, population density and satellite imagery. Together, this collection can help extend our understanding of urban environmental processes through observational synthesis studies or in the evaluation of land surface environmental models in a wide range of urban settings.

## 1 Background

Tower mounted instruments allow the measurement of land-atmosphere fluxes (e.g. energy, momentum, water, carbon) and local meteorological conditions. These observations are one of the fundamental ways of improving both our understanding and ability to predict biogeophysical and weather-related processes at local scales. Regional and global networks of flux tower sites have helped extend our knowledge of ecosystem and climate science (Novick et al., 2018; Beringer et al., 2016; Yamamoto et al., 2005; Valentini, 2003). Over the last 25 years networks such as FLUXNET have progressively increased access to flux data through open-source collections (Pastorello et al., 2020), extending the reach and impact of individual site observations through synthesis studies (Baldocchi, 2020) and multi-site environmental modelling and model evaluation projects (Best et al., 2015; Ukkola et al., 2022). However, with few urban sites included, urban areas have not benefited from the improved understanding or more extensive model evaluations that these collections can facilitate.

Urban areas are unique ecosystems, distinct from natural or rural landscapes. First, most people live in cities (UN, 2018) and infrastructure is concentrated within them. Therefore, climate-related health and economic impacts fall disproportionately within urban areas. Second, urban infrastructure (e.g. buildings and roads) along with transient human activities (e.g. energy consumption and irrigation) fundamentally alter surface energy, water and mass exchanges with the atmosphere, modifying local and larger-scale environmental conditions (Oke et al., 2017). Third, as built environments, urban areas are uniquely capable of actively mitigating and adapting to climate change.

Establishing and maintaining long term flux sites in cities is particularly challenging because of the rarity of appropriate sites with homogenous fetch, the difficulty in gaining approval to access existing towers (e.g. for telecommunications), the cost of constructing tall towers over an aerodynamically rough surface, and extremely limited long-term funding opportunities (Arnfield, 2003; Grimmond, 2006; Velasco and Roth, 2010; Feigenwinter et al., 2012; Grimmond and Ward, 2021). Thus, despite the diversity and importance of urban areas across the globe, urban flux tower data are relatively scarce, generally of



short duration and rarely open source. Databases identifying urban observational programmes exist (e.g. the Urban Flux
Network (Grimmond and Christen, 2012)), however urban flux tower datasets have not previously been brought together into
a harmonised, gap-filled, open access collection.

We bring together quality-controlled data from 20 urban sites in an open collection that includes 50 observation-years (Lipson
et al., 2022). The sites are chosen to be diverse in both regional climates and urban characteristics. As evaluating land surface
models is one key application for these data, we create continuous forcing data sets (i.e. with incoming radiation fluxes and
other meteorological data) that are gap filled using site specific, bias corrected reanalysis data. Observations are also prepended
with 10 years of site-specific reanalysis-derived meteorological data to allow modelled soil moisture and other conditions to
equilibrate with local climate conditions during model spin up.

Along with the meteorological data, site characteristics and metadata are provided in a common format. The metadata includes
tower location, land cover fractions, building heights and morphology, aerodynamic roughness parameter estimates, population
density, estimated anthropogenic heat fluxes, site photos and satellite imagery. Together, this collection can help extend our
understanding of urban environmental processes through observational synthesis studies, or in the evaluation of land surface
environmental models in different urban settings.

## 2 Methods

### 2.1 Site selection

The initial motivation for collating these flux tower and site data is for use in the Urban-PLUMBER multi-site model evaluation
project, currently underway (Urban-PLUMBER: A multi-site model evaluation project for urban areas - Project Home, 2021).
Urban-PLUMBER draws on methods from the first international urban land surface model comparison (Grimmond et al.,
2010, 2011) and the Protocol for the Analysis of Land Surface Models Benchmarking Evaluation Project (PLUMBER (Best
et al., 2015)). The latter evaluated land surface models in non-urban (vegetated) areas, while Urban-PLUMBER evaluates land
surface models at 20 urban sites (Fig. 1, Table 1).

There is a two-fold use of these observational data in the model evaluation of Urban-PLUMBER:

1. To provide local-scale meteorological input forcing to drive land surface models
2. To evaluate the performance of models, primarily assessing the local-scale exchange of radiant and turbulent heat
   fluxes between the surface and lower atmosphere

With these objectives in mind, the following criteria are used to select flux tower sites:
- Appropriately sited for neighbourhood-scale conditions - i.e. within the inertial sub-layer, typically 2-5 times above
  the average building height and with relatively homogenous fetch (Grimmond, 2006; Barlow, 2014; Grimmond and
  Ward, 2021)





- Requested observations available at 30- or 60-minute resolution (Table 2)
- Local site characteristics available for description and configuring models
- A preference for longer datasets (as this allows seasonal and inter-annual variability to be included)
- Collectively represent a diverse range of site characteristics and climates

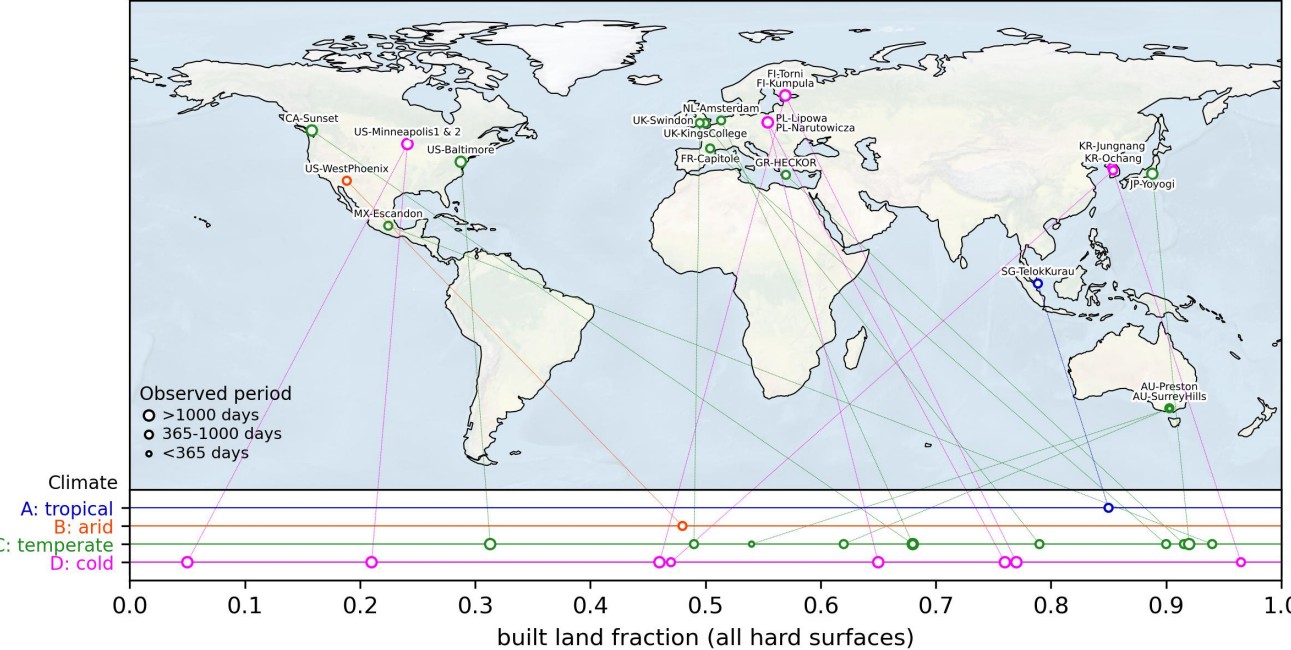

**Figure 1**: Location of flux tower sites in this collection. Each site Köppen-Geiger climate classification (Beck et al., 2018) and the built land
fraction around the tower are indicated at the bottom of the figure.

**Table 1:** Site location and included observation (focus) period. Data providers may have longer observation periods available than are in this collection. Resolution is 30 minutes (or 60 minutes if denoted by *). All periods in universal time coordinated (UTC). US-Minneapolis data are split based on wind direction and fetch (Section 4.5).

| Sitename | City | Country | Observed period | Latitude | Longitude | References |
|---|---|---|---|---|---|---|
| AU-Preston | Melbourne | Australia | Aug 2003 – Nov 2004 | -37.7306 | 145.0145 | (Coutts et al., 2007a, b) |
| AU-SurreyHills | Melbourne | Australia | Feb 2004 – Jul 2004 | -37.8265 | 145.099 | (Coutts et al., 2007a, b) |
| CA-Sunset | Vancouver | Canada | Jan 2012 – Dec 2016 | 49.2261 | -123.078 | (Christen et al., 2011; Crawford and Christen, 2015) |
| FI-Kumpula | Helsinki | Finland | Dec 2010 – Dec 2013 | 60.2028 | 24.9611 | (Karsisto et al., 2016) |
| FI-Torni | Helsinki | Finland | Dec 2010 – Dec 2013 | 60.1678 | 24.9387 | (Järvi et al., 2018; Nordbo et al., 2013) |
| FR-Capitole | Toulouse | France | Feb 2004 – Mar 2005 | 43.6035 | 1.4454 | (Masson et al., 2008; Goret et al., 2019) |
| GR-HECKOR | Heraklion | Greece | Jun 2019 – Jun 2020 | 35.3361 | 25.1328 | (Stagakis et al., 2019) |
| JP-Yoyogi | Tokyo | Japan | Mar 2016 – Mar 2020* | 35.6645 | 139.6845 | (Hirano et al., 2015; Ishidoya et al., 2020) |
| KR-Jungnang | Seoul | South Korea | Jan 2017 – Apr 2019 | 37.5907 | 127.0794 | (Jo et al., n.d.; Hong et al., 2020) |
| KR-Ochang | Ochang | South Korea | Jun 2015 – Jul 2017 | 36.7197 | 127.4344 | (Hong et al., 2019, 2020) |
| MX-Escandon | Mexico City | Mexico | Jun 2011 – Sep 2012 | 19.4042 | -99.1761 | (Velasco et al., 2011, 2014) |
| NL-Amsterdam | Amsterdam | Netherlands | Jan 2019 – Oct 2020 | 52.3665 | 4.8929 | - |





| | | | | | | |
|---|---|---|---|---|---|---|
| PL-Lipowa | Łódź | Poland | Jan 2008 – Dec 2012* | 51.7625 | 19.4453 | (Fortuniak et al., 2013; Pawlak et al., 2011) |
| PL-Narutowicza | Łódź | Poland | Jan 2008 – Dec 2012* | 51.7733 | 19.4811 | (Fortuniak et al., 2013, 2006) |
| SG-TelokKurau06 | Singapore | Singapore | Apr 2006 – Mar 2007 | 1.3143 | 103.9112 | (Roth et al., 2017) |
| UK-KingsCollege | London | UK | Apr 2012 – Jan 2014 | 51.5118 | -0.1167 | (Bjorkegren et al., 2015; Kotthaus and Grimmond, 2014a, b) |
| UK-Swindon | Swindon | UK | May 2011 – Apr 2013 | 51.5846 | -1.7981 | (Ward et al., 2013) |
| US-Baltimore | Baltimore | USA | Jan 2002 – Jan 2007* | 39.4128 | -76.5215 | (Crawford et al., 2011) |
| US-Minneapolis | Minneapolis | USA | Jun 2006 – May 2009 | 44.9984 | -93.1884 | (Peters et al., 2011; Menzer and McFadden, 2017) |
| US-WestPhoenix | Phoenix | USA | Dec 2011 – Jan 2013 | 44.9984 | -93.1884 | (Chow, 2017; Chow et al., 2014) |

Potential sites identified from published site lists (Grimmond and Christen, 2012; Oke et al., 2017) and open calls for data (e.g.
community newsletters (Lipson et al., 2020a), international conferences (Lipson et al., 2020b, c) and social media professional networks). We deemed 20 sites sufficient for the evaluation project (Table 1), together covering a 50 site-years. Included sites have built fractions (i.e. plan area fraction of all impervious surfaces including roofs, roads, other paving *etc.*) from 0.05 to 0.965, and are located in four major Köppen-Geiger (Beck et al., 2018) climate classes (Fig. 1). Eleven sites are in temperate climates, eight in cold (or continental) climates, and one in each of tropical and arid climates.

Sites are reasonably distributed across mean temperature and precipitation for global urban locations, but gaps remain, particularly in warm, wet and very cold climates (Fig. 2). Some urban flux observations in understudied regions were not included (e.g., Ouagadougou (Offerle et al., 2005), São Paulo (Ferreira et al., 2013), Guangzhou (Shi et al., 2019), Beijing (Dou et al., 2019)) because they do not meet the model evaluation project needs because of the relatively short observed periods for the available data. These regions and climates have large urban populations with significant environmental challenges and
have few urban flux tower sites compared with northern hemisphere temperate or continental locations (Grimmond, 2006; Roth et al., 2017). Understudied regions and climates should be included in future collections when appropriate time series become available.



**Figure 2:** Climatology of included sites compared with more than 70,000 global urban areas. Mean temperature and annual precipitation at the 20 tower sites (red, truncated site name, Table 1) from tower observations; global urban locations (grey) from ERA5 surface data (Hersbach et al., 2020, 2018) (2000 – 2010) from grid nearest to locations identified in the Global Rural-Urban Mapping Project (GRUMP) (Center for International Earth Science Information Network - CIESIN - Columbia University et al., 2017). Locations with rainfall above 3000 mm year-1 (1.3% of locations), and mean temperature below -3°C (0.2 % of locations) are not shown.




## 2.2 Flux tower data

The observed data are provided in 30- or 60-minute periods (Table 1), processed from high-frequency samples by individual observing groups. In the harmonized collection time stamps are in coordinated universal time (UTC) indicating the end of the measurement period. Variables name and units use ALMA conventions: (Assistance for Land-surface Modelling Activities), a format used in previous land surface model comparisons.

Data are cleaned (Sect 2.3: *Quality control*), gap filled (Section 2.4) and prepended with data derived from ERA5 (Section 2.5:) after site-specific corrections (Section 2.6). An example of final prepended and gap-filled data is shown in Figure 3 for one site (UK-KingsCollege). Plots for all other variables and sites are also available in the collection (Lipson et al., 2022).

  Data are split into forcing and analysis variable sets (Table 2) to allow the forcing variables to be provided to modelling groups as input to run their models. The withheld analysis data are used by the coordinating group to assess the model outputs.

Some additional observed variables (Table 3) have, where practical, been included in the datasets after passing through the quality control steps. Missing forcing variables are obtained using bias-corrected reanalysis data (Section 2.4). No gap filling is applied to analysis data or additional variables.




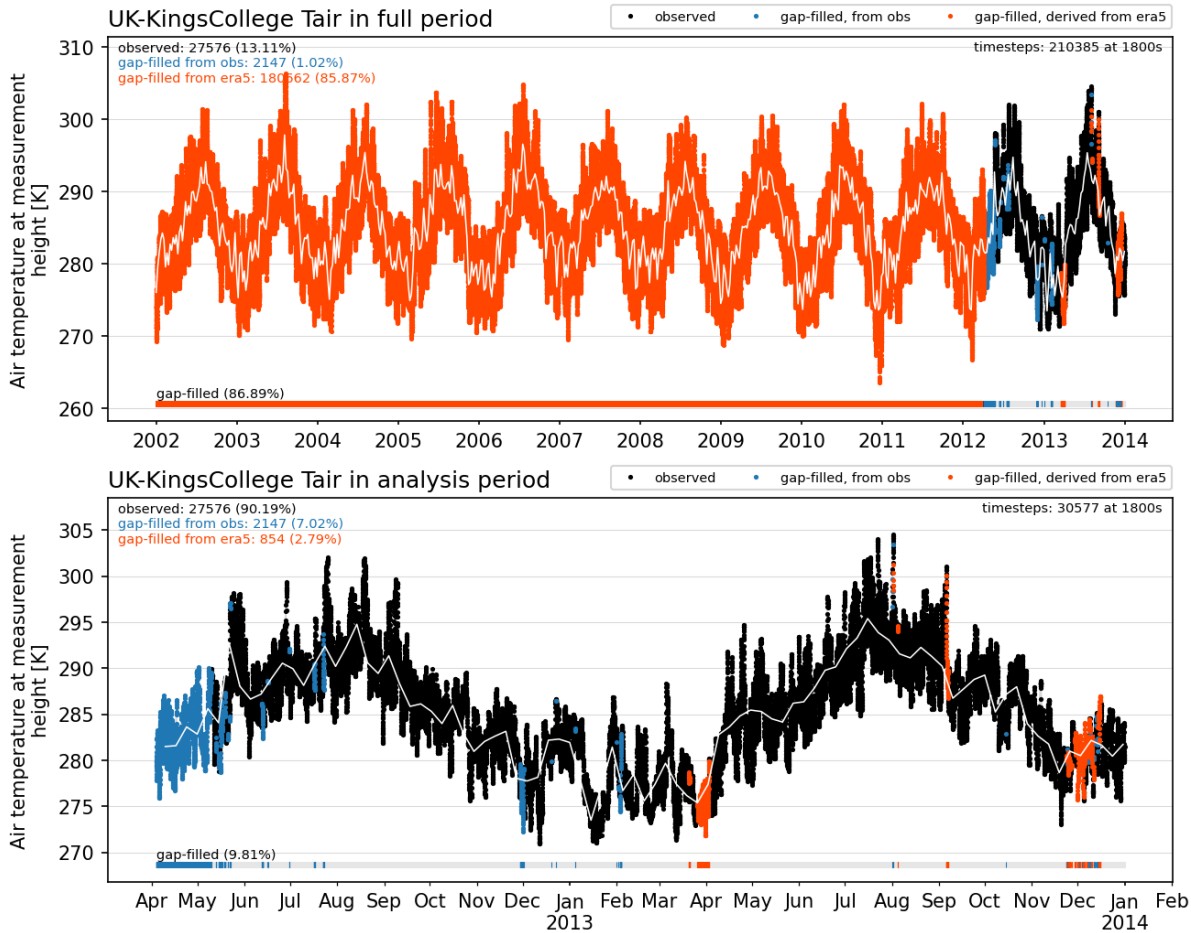

**Figure 3:** Forcing timeseries with gap filling. Example shown for air temperature (Tair) at Kings College, London (UK-KingsCollege); (a) full forcing period, including 10 years of ERA5-derived data (red) prior to observations (black) used for model spin up, and (b) focus period used for analysis. Gaps are first filled from nearby tower measurements where available, and short gaps (<=2 hours) are linearly interpolated (blue). Remaining gaps are filled using the ERA5-derived timeseries which is seasonally and diurnally bias corrected using site observations. White lines show seven day mean values. Similar plots are available for other sites within the site data collection (Lipson et al., 2022).

**Table 2:** Forcing and analysis flux tower data variables. Short name description, units and positive direction use ALMA data conventions. Mean annual estimates of anthropogenic heat flux are included as site metadata. Note ground heat flux (Qg) is the heat flux into soil rather than total storage heat flux which is difficult to measure in urban areas (Grimmond and Oke, 1999).

| Variable | Description | Units | Positive direction | ERA5 bias correction |
|---|---|---|---|---|
| **Forcing data** | | | | |
| SWdown | Downward shortwave radiation | W m$^{-2}$ | Downward | none |
| LWdown | Downward longwave radiation | W m$^{-2}$ | Downward | hourly and daily |
| Tair | Air temperature | K | - | hourly and daily |
| Qair | Specific humidity | kg kg$^{-1}$ | - | hourly and daily |
| PSurf | Station air pressure | Pa | - | hourly and daily |
| Wind_N | Northward wind component | m s$^{-1}$ | Northward | logarithmic law |
| Wind_E | Eastward wind component | m s$^{-1}$ | Eastward | logarithmic law |





| Variable | Description | Units | Positive direction | ERA5 bias correction |
|---|---|---|---|---|
| **Forcing data** | | | | |
| Rainf | Rainfall rate | kg m$^{-2}$ s$^{-1}$ | Downward | long term precipitation |
| Snowf | Snowfall rate | kg m$^{-2}$ s$^{-1}$ | Downward | long term precipitation |
| **Analysis data** | | | | |
| SWup | Upward shortwave radiation | W m$^{-2}$ | Upward | none |
| LWup | Upward longwave radiation | W m$^{-2}$ | Upward | none |
| Qle | Latent heat flux | W m$^{-2}$ | Upward | none |
| Qh | Sensible heat flux | W m$^{-2}$ | Upward | none |
| **Additional data (optional)** | | | | |
| Qg | Ground heat flux into soil | W m$^{-2}$ | Downward | none |
| Qtau | Momentum flux | N m$^{-2}$ | Downward | none |
| Tair2m | Near surface air temperature (2 m) | K | - | none |
| SoilTemp | Soil temperature (depth in metadata) | K | - | none |

**Table 3:** Site climate classification, missing and additional variables. Climate classification from Köppen-Geiger global dataset (Beck et al., 2018). Table 2 gives variable definitions. Note: as MX-Escandon LWdown data are unavailable during the 2011-2012 focus period, but were available in 2006, this earlier period is used to determine bias correction for ERA5 LWdown data.

| Sitename | Class | Climate description | Missing variables | Additional variables |
|---|---|---|---|---|
| AU-Preston | Cfb | Temperate, no dry season, warm summer | Snowf | Qtau |
| AU-SurreyHills | Cfb | Temperate, no dry season, warm summer | Snowf | Qtau |
| CA-Sunset | Csb | Temperate, dry summer, warm summer | Snowf | Qtau, SoilTemp |
| FI-Kumpula | Dfb | Cold, no dry season, warm summer | Snowf | |
| FI-Torni | Dfb | Cold, no dry season, warm summer | Snowf | |
| FR-Capitole | Cfa | Temperate, no dry season, hot summer | Snowf | Qtau |
| GR-HECKOR | Csa | Temperate, dry summer, hot summer | Snowf | Qtau |
| JP-Yoyogi | Cfa | Temperate, no dry season, hot summer | | |
| KR-Jungnang | Dwa | Cold, dry winter, hot summer | Snowf | |
| KR-Ochang | Dwa | Cold, dry winter, hot summer | Snowf | |
| MX-Escandon | Cwb | Temperate, dry winter, warm summer | Snowf, LWdown* | Qtau |
| NL-Amsterdam | Cfb | Temperate, no dry season, warm summer | Snowf | Qtau |
| PL-Lipowa | Dfb | Cold, no dry season, warm summer | Snowf | |
| PL-Narutowicza | Dfb | Cold, no dry season, warm summer | Snowf | |
| SG-TelokKurau06 | Af | Tropical, rainforest | Snowf | |
| UK-KingsCollege | Cfb | Temperate, no dry season, warm summer | Snowf | Qtau |
| UK-Swindon | Cfb | Temperate, no dry season, warm summer | Snowf | Qtau |
| US-Baltimore | Cfa | Temperate, no dry season, hot summer | Snowf | Qtau, SoilTemp |
| US-Minneapolis | Dfa | Cold, no dry season, hot summer | Snowf | Qtau, SoilTemp, Qg |
| US-WestPhoenix | BWh | Arid, desert, hot | Snowf | Qtau, SoilTemp |

## 2.3 Quality control and assurance

For each site the 30- or 60-minute variables are calculated by data providers from high-frequency samples after applying their own quality control measures (e.g. Aubinet et al., 2012; Feigenwinter et al., 2012; Kotthaus and Grimmond, 2012; Vitale et al., 2020). The harmonised collection consists of the data retained after undergoing five additional quality control steps, in this order:





1. **Out-of-range**: removal of unphysical values (e.g. negative shortwave radiation) using the ALMA expected range protocol (Bowling and Polcher, 2001).

2. **Night**: nocturnal shortwave radiation set to zero, based on civil twilight (when the sun is 6° below the horizon (Forsythe et al., 1995)).

3. **Constant**: four or more timesteps with identical values (excluding zero values for shortwave radiation, rainfall and snowfall) are removed as suspicious.

4. **Outlier**: values outside ±4 standard deviations for each hour in a rolling 30-day window (to account for diurnal and seasonal variations) removed. Repeat with a larger tolerance (± 5 standard deviations) until no outliers remain (Schmid et al., 2000). The outlier test is not applied to precipitation.

5. **Visual**: remaining suspect readings are removed manually via visual inspection.

These steps are undertaken in the processing script *qc_observations.py* (see Sect 5*: Code availability*), including periods identified through visual inspection (21 instances across all data). Data removed through quality control are indicated in plots of each variable at each site included in the data collection (Lipson et al., 2022). Unaltered (raw) data are also included in the collection.

Communication, or human errors, also have the potential to degrade or invalidate data (Menard et al., 2021). As part of quality assurance, project coordinators prepared an observational data protocol (Lipson et al., 2021) to explicitly set out requirements for data providers prior to submission of their data. The protocol documented instrument siting requirements, variables and data formats, dataset length and resolution, necessary site characteristic information and metadata, as well as the expectations for data handling, use and authorship. On receiving data, coordinators undertook further checks and identified errors that were not be picked up by automated quality control. Identified errors included mislabelled variables and metadata, inconsistent timestamps and unit discrepancies. Many of the errors were identified by comparing provided data with secondary sources such as ERA5, nearby meteorological stations or previous publications. Errors were corrected collaboratively with data providers, some leading to corrections in primary data sources.

**2.4 Gap-filling**

Three gap filling methods are used to create a continuous dataset for forcing variables, in this order:

- contemporaneous and nearby flux tower or weather observing sites (where available from data providers)
- small gaps (≤ 2 hours) are linearly interpolated from the adjoining observations
- larger gaps and a 10-year spin-up period are filled with bias corrected ERA5 data (Section 2.6).

As only one site provided observed snowfall rate (JP-Yoyogi), ERA5 snowfall rates are used for all periods at other sites. At those sites the additional water equivalent from ERA5 snowfall is removed from subsequent observed rainfall until mass balance of observed total precipitation is achieved. This corrects melting snow being recorded as rainfall.



**2.5 ERA5 reanalysis data**

The ERA5 reanalysis product (Hersbach et al., 2020) assimilates global satellite, atmospheric and ground-based observations
to constrain numerical weather prediction simulations, producing global output at 0.25° spatial and hourly temporal resolutions
from 1979 to the present. It is therefore useful as a globally consistent and accessible source of meteorological data across
space and time. ERA5, and its lower resolution predecessor ERA-Interim (Dee et al., 2011), have been used extensively to
provide meteorological forcing data to drive land surface models and gap fill flux tower observations (Vuichard and Papale,
2015; Kokkonen et al., 2018; Pastorello et al., 2020; Ukkola et al., 2017, 2022).

The ERA5 hourly single level (Hersbach et al., 2018) dataset (retrieved from NCI Australia (Druken, 2020)) is used for gap-
filling missing observations within the focus periods (Table 1) and for the 10-year model spin-up period. However, combining
ERA5 data directly with urban flux tower observations has several deficiencies.

Grid-scale ERA5 data are not directly compatible with point-scale urban flux tower observations. This incompatibility is three-
fold:
1.  **Horizontally**: The ERA5 grid cell area (of order 30 x 30 km$^2$) does not match the flux footprint from tower observations
(of order 1 km$^2$). The ERA5 surface characteristics, including elevation, are based on an average description for the grid
which may differ from surface characteristics around the observing tower, particularly in coastal or mountainous regions
(Martens et al., 2020) in which many cities are located.
2.  **Vertically**: ERA5 provide near-surface variables (2 or 10 m above ground level), aligning with World Meteorological
Organization (WMO) guidelines for standard regional observations taken over short grass (World Meteorological
Organization, 2008). As the urban roughness elements (e.g. buildings) are much taller than grass, instruments are mounted
on towers at heights greater than 2 – 5 times average building height in order to be located within the inertial sub layer or
constant flux layer (Velasco and Roth, 2010; Barlow, 2014; Grimmond and Ward, 2021).
3.  **Land surface**: As the current operational ERA5 modelling systems do not include an urban land surface scheme
(Boussetta et al., 2013; McNorton et al., 2021), other land types (grass, crops, shrubs, trees etc.) are to characterise the
grid cell (Table 4). Urban land surfaces are well known to alter local meteorological conditions (Oke et al., 2017), therefore
ERA5 output will likely differ from locally observed conditions.

Outside of cities there are known diurnal and seasonal biases between the ERA5 near-surface variables and observations
(Haiden et al., 2018; Betts et al., 2019; Nogueira, 2020; Martens et al., 2020; Jiang et al., 2021). These biases are an outcome
of simplifying assumptions made in model parameterisations and inadequacies of modelling frameworks in general (Cucchi et
al., 2020). Various approaches to reduce ERA5 biases in non-urban areas have been proposed. For example, the Water and
Global Change (WATCH) Forcing Data (WFD) project use gridded observations to bias-correct ERA-Interim data (Weedon
et al., 2011), and more recently ERA5 data, creating the global WFDE5 dataset for impact studies (Cucchi et al., 2020). WFDE5





relies on the Climate Research Unit (CRU) monthly timeseries of gridded observations with resolution courser than ERA5
(New et al., 1999), requiring ERA5 to be regridded to a lower resolution. This may reduce the representativeness of the ERA5
data, particularly in heterogenous or complex terrain.

Alternatively, local observations can be used to bias-correct ERA data, e.g. the linear regression corrections using tower
observations applied to FLUXNET datasets (Vuichard and Papale, 2015; Pastorello et al., 2020). However, linear methods
neither conserve the variability of observations (Vuichard and Papale, 2015) (Section 3), nor can they correct diurnal timing
differences within ERA5 data (e.g. out of phase from urban temporal profiles, which are typically delayed compared with non-
urban surfaces used in ERA5, Figure 4, Table 4).

To account for these deficiencies, we develop a novel set of methods to bias correct ERA5 data to better represent observed
urban conditions (Section 2.6).

**Table 4**: ERA5 surface cover information for grid cells in which the urban flux towers are located (Table 1). The land surface model used
for ERA5 simulations does not account for urbanised land surfaces, nor vegetation types typical at urban sites. ERA5 surface roughness
values are indicative as they vary slightly through time. Effective roughness is a correction accounting for observed urban mean wind speeds.

| Site | ERA5 low vegetation | ERA5 high vegetation | low vegetation fraction | high vegetation fraction | lake (or sea) fraction | bare soil fraction | ERA5 surface roughness [m] | effective roughness [m] |
|---|---|---|---|---|---|---|---|---|
| AU-Preston | tall grass | interrupted forest | 0.484 | 0.407 | 0.088 | 0.021 | 0.514 | 0.289 |
| AU-SurreyHills | tall grass | interrupted forest | 0.484 | 0.407 | 0.088 | 0.021 | 0.514 | 0.368 |
| CA-Sunset | crops, mixed farming | evergreen needleleaf trees | 0.205 | 0.723 | 0.071 | 0.000 | 1.077 | 1.508 |
| FI-Kumpula | crops, mixed farming | evergreen needleleaf trees | 0.296 | 0.352 | 0.137 | 0.215 | 0.708 | 0.703 |
| FI-Torni | crops, mixed farming | evergreen needleleaf trees | 0.296 | 0.352 | 0.137 | 0.215 | 0.708 | 0.424 |
| FR-Capitole | crops, mixed farming | interrupted forest | 0.920 | 0.050 | 0.004 | 0.025 | 0.291 | 0.519 |
| GR-HECKOR | crops, mixed farming | interrupted forest | 0.172 | 0.463 | 0.158 | 0.207 | 0.505 | 1.187 |
| JP-Yoyogi | semidesert | no vegetation recorded | 0.943 | 0.000 | 0.010 | 0.047 | 0.015 | 0.649 |
| KR-Jungnang | crops, mixed farming | evergreen needleleaf trees | 0.781 | 0.168 | 0.051 | 0.000 | 0.516 | 0.074 |
| KR-Ochang | irrigated crops | interrupted forest | 0.281 | 0.716 | 0.003 | 0.000 | 0.844 | 0.181 |
| MX-Escandon | evergreen shrubs | mixed forest/woodland | 0.743 | 0.216 | 0.006 | 0.035 | 0.404 | 0.229 |
| NL-Amsterdam | crops, mixed farming | interrupted forest | 0.867 | 0.061 | 0.056 | 0.015 | 0.248 | 0.254 |
| PL-Lipowa | crops, mixed farming | interrupted forest | 0.855 | 0.144 | 0.001 | 0.000 | 0.250 | 0.306 |
| PL-Narutowicza | crops, mixed farming | interrupted forest | 0.855 | 0.144 | 0.001 | 0.000 | 0.250 | 0.558 |
| SG-TelokKurau06 | irrigated crops | interrupted forest | 0.905 | 0.021 | 0.074 | 0.000 | 0.335 | 0.309 |
| UK-KingsCollege | crops, mixed farming | interrupted forest | 0.609 | 0.372 | 0.020 | 0.000 | 0.504 | 0.315 |
| UK-Swindon | crops, mixed farming | interrupted forest | 0.727 | 0.251 | 0.001 | 0.021 | 0.397 | 0.146 |
| US-Baltimore | crops, mixed farming | deciduous broadleaf trees | 0.044 | 0.908 | 0.048 | 0.000 | 1.675 | 1.076 |
| US-Minneapolis1 | crops, mixed farming | interrupted forest | 0.228 | 0.706 | 0.059 | 0.006 | 0.814 | 0.242 |
| US-Minneapolis2 | crops, mixed farming | interrupted forest | 0.228 | 0.706 | 0.059 | 0.006 | 0.814 | 0.406 |
| US-WestPhoenix | semidesert | evergreen needleleaf trees | 0.949 | 0.051 | 0.000 | 0.000 | 0.084 | 0.404 |

## 2.6 Bias-correction methods

Bias-correction approaches used in the collection depend on the forcing variable (Table 2) and are described below.



### 2.6.1 Hourly and daily corrections

For incoming longwave radiation, air temperature, specific humidity and air pressure, the mean bias between ERA5 and local flux tower observations are calculated for each hour ($h$) and each day of a year ($D$) in a 60-day rolling window of a representative year (Fig. 4a). The calculated bias $\eta_{bias}(D, h)$ is subtracted from the complete ERA5 timeseries $\eta_{ERA5}(t)$ to create a new corrected timeseries:

$$\eta(t) = \eta_{ERA5}(t) - \eta_{bias}(D, h). \tag{1}$$

The ERA5 data is from the grid nearest the observation site with at least 50% land. The resulting corrected timeseries (e.g. Fig. 4b) is used for gap filling site observations and for the spin up period. The subroutine *rolling_hourly_bias_correction* in the file *pipeline_functions.py* (Section 5) undertakes corrections with the following steps:

1.  If observations have a 30-min resolution, average to 60-min to match ERA5 periods
2.  Remove ERA5 data for time periods where site observations are missing
3.  Calculate mean for each hour with both data sets to create a 'representative' year of 366 days
4.  Extend to a three-year period by duplication to provide smoother transitions at year end
5.  Calculate hourly means in a 60-day rolling window across the repeating timeseries, excluding data in windows with less than 30 observations. Repeat mean calculation for greater smoothing.
6.  Calculate a timeseries of the bias between observed and ERA5 rolling means
7.  Fill gaps in the bias timeseries by linear interpolation through each hour separately
8.  Remove first and last year in the bias timeseries, using only the central year to bias correct each hour in the original ERA5 timeseries

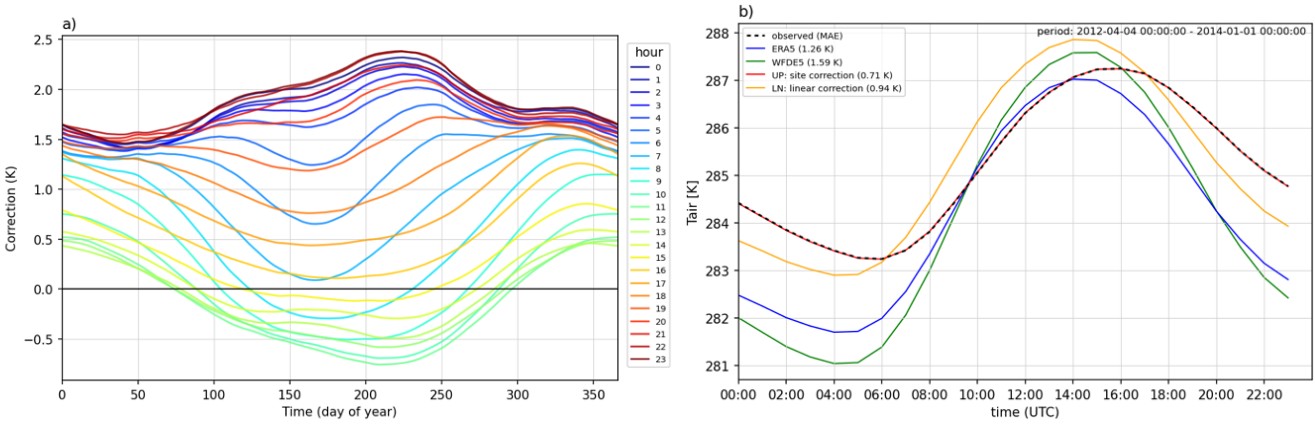

**Figure 4:** Urban-PLUMBER reanalysis bias correction methods. Demonstrated using air temperature (Tair) for the grid containing the
King's College London site (UK-KingsCollege). (a) hourly (colour) bias calculated for each day of a 'representative' year and applied to entire ERA5 timeseries; (b) diurnal hourly mean Urban-PLUMBER correction (UP, red), observations (black), original ERA5 data (blue),



WFDE5 bias corrected data (green) and linear bias correction method used in FLUXNET (LN, yellow). Our new UP method has smaller mean absolute errors (MAE) overall, and can correct both pattern and phase errors of ERA5 (Section 3).

**2.6.2 Logarithmic wind profile correction**

Wind speed differences between ERA5 and site observations can result from errors in modelled synoptic-scale speeds, differences in representative heights, and differences in surface aerodynamic properties like roughness and displacement height. To correct bias and maintain standard deviations of the wind components (U,V) of observations at sensor height ($z_{site}$), the following correction to ERA5 data is undertaken assuming both a logarithmic wind profile and neutral conditions (Goret et al., 2019):

$$u_{corr} = u_{ERA5} \frac{\ln\left(\frac{z_{site}-d_{site}}{z_{0,site}}\right)}{\ln\left(\frac{z_{grid}-d_{grid}}{z_{0,grid}}\right)}, \qquad (2)$$

where $u_{corr}$ is the corrected wind speed at $z_{site}$. The ERA5 wind ($u_{ERA5}$) at 10 m ($z_{grid}$) is used with the site surface roughness ($z_{0,site}$) and displacement height ($d_{site}$), grid roughness ($z_{0,grid}$) (Table 4) while assuming grid displacement height ($d_{grid}$) is zero for simplicity. If the resulting mean value of $u_{corr}$ differs from observed mean value by more than 0.01 m s$^{-1}$, then $z_{0,grid}$ is iteratively adapted until this threshold accuracy in mean wind speed is achieved. Derived $z_{0,grid}$ values are given in Table
4 (last column). Note this approach ignores seasonal effects from vegetation phenology and directional effects but ensures mean wind speeds are appropriate at the urban $z_{site}$ while conserving variability within the ERA5 derived wind data.

**2.6.3 Long term precipitation correction**

Total precipitation is an important variable in urban land surface models because of the effect on soil moisture which evolves over multi-year periods (Best and Grimmond, 2014). Most of the observational datasets included are not long enough to capture
interannual variations. We therefore use longer term total precipitation (*P*) from nearby stations from the Global Historical Climatology Network - Daily (GHCND) (Menne et al., 2012) over a 10 year period to correct ERA5 rain and snow fluxes ($\phi$) at each timestep:

$$\phi_{corr} = \frac{\sum_{i=1}^{10\,yrs} P_{GHCND}}{\sum_{i=1}^{10\,yrs} P_{ERA5}} \phi_{ERA5}(t). \qquad (3)$$

Gaps in the nearest GHCND station data are progressively filled by the next nearest station until no gaps are present. If gaps
could not be filled with GHNCD stations within 2° of latitude and longitude from the flux tower, and no alternative records are found (e.g. from national meteorological and hydrological services), then ERA5 rates are used unadjusted (*viz.*, KR-Ochang and KR-Jungnang). This assumes precipitation occurs on the same dates and at the same times in ERA5 and observed datasets, which may become less valid under increasingly convective conditions.



### 2.6.4 Linear bias correction

The FLUXNET2015 collection of 212 flux tower sites (Pastorello et al., 2020) bias correction method uses linear regression between site observations and reanalysis data to derive one slope ($s$) and intercept ($b$) per site, hence 'unbiasing' all ERA timesteps ($i$) :

$$LN_i = s \cdot ERA5_i + b. \tag{4}$$

Following FLUXNET2015, global radiation and wind fields are assigned an intercept of zero, and precipitation is not linearly
modified (Vuichard and Papale, 2015). FLUXNET2015 use the coarser resolution ERA-Interim (spatial: 0.5° cf. 0.25°; temporal 3-h cf. 1-h) than ERA5. In this evaluation we use ERA5, which is found to be a consistent improvement over ERA-Interim (Albergel et al., 2018). However, after assessment we chose not to use a linear method (LN) for correcting variables in this collection. Its description is retained here for comparison purposes (Section 3).

### 3 Gap-filling evaluation

Site observations are quality controlled by individual data providers and collectively for this project (Section 2.3). The observed data required for forcing land surface models are then gap filled using a novel method of bias correcting reanalysis data. In this section, four methods which draw on ERA5 data are evaluated:

1.   **ERA5**: nearest land based 0.25° resolution ERA5 (Hersbach et al., 2018, p.5) grid without bias correction
2.   **W5**: nearest WFDE5 (Cucchi et al., 2020, p.5) grid (which uses bias correction from 0.5° CRU monthly gridded
observations)
3.   **UP**: the Urban-PLUMBER methods described here (using site observations for bias correction)
4.   **LN**: linear methods based on FLUXNET2015 (Vuichard and Papale, 2015; Pastorello et al., 2020) (using site observations for bias correction)

To evaluate the methods available for gap filling, quality-controlled tower site observations ($O_i$) are used to assess the
calculated value ($\eta$) at timestep $i$ using three metrics:

a)   Mean bias error (MBE): $\frac{\sum_{i=1}^{n} \eta_i - O_i}{n}$

b)   Mean absolute error (MAE): $\frac{\sum_{i=1}^{n} |\eta_i - O_i|}{n}$

c)   Normalised standard deviation (nSD): $\frac{\sum_{i=1}^{n} (\eta_i - \bar{\eta})^2}{n-1} \Big/ \frac{\sum_{i=1}^{n} (O_i - \bar{O})^2}{n-1}$

where $\bar{O}$ and $\bar{\eta}$ are time-averaged over $n$ data points. The time stamps for each variable are made consistent between the data
sources: ERA5 are hourly time ending data (Hersbach et al., 2018, p.5); 60-min site observations are hour ending; 30-min site





observations are converted to 60-min time ending by averaging; whereas as the WFDE5 SWdown, LWdown and Rainf are natively 60-min time beginning (Cucchi et al., 2020, p.5) they are shifted forward to match the time ending timestamps; and WFDE5 Tair, Qair, Psurf and Wind are instantaneous samples on the hour so their time stamp remains unchanged.

To summarise each metric for the 20 sites, we use boxplots (Fig. 5) for the seven forcing variables. The evaluation inherently
considers the net differences associated with both the spatial (vertical and surface cover) differences and errors (model and observation) from the two datasets. Uncorrected ERA5 (blue, Fig. 5) biases are generally negative for Tair, LWdown and Wind, and generally positive for Qair and SWdown. These biases can be partly explained by the ERA5 framework not including an urban surface model. For example, the well-documented warmer air temperature in cities (urban heat island) are not modelled in ERA5 because natural land surfaces are assumed in simulations (Table 4), although ERA5 can include an
urban signal if the data assimilated are from within urban areas (Tang et al., 2021). Qair shows a general positive bias as evapotranspiration will be overestimated in ERA5 without an urban land surface representation. Likewise, ERA5 SWdown are overestimated and LWdown underestimated possibly because urban air pollution effects are not included (Oke, 1988).

Other discrepancies between ERA5 data and site data arise from elevation differences in height above sea level (asl) of the ERA5 grid and site. For example, the MX-Escandon tower in Mexico City measurement height is 2277 m asl, whereas the
ERA5 grid cell is assigned a surface elevation of 2540 m asl because the cell includes nearby mountains. This 263 m difference causes a negative bias to Psurf of 2594 Pa and contributes to a Tair difference of -2.15 K. Additionally, orographic uplift increases the grid cell rainfall, leading to positive ERA5 bias of 2.25 x $10^{-5}$ kg m$^{-2}$ s$^{-1}$ (+710 mm year$^{-1}$) compared with the MX-Escandon site observations. The ERA5 rainfall bias is even more pronounced at CA-Sunset in Vancouver, Canada, with a +1178 mm year$^{-1}$ bias. These results are consistent with other studies highlighting discrepancies between reanalysis and local
data in mountainous regions (Kokkonen et al., 2018).



**Figure 5:** Evaluation of bias correction methods. Four methods (colour) to create gap filled observed time series data: ERA5 (blue), WFDE5 (W5, green), linear debiasing (LN, orange), UP (red, this study) using (row 1) mean bias error, (row 2) mean absolute error, (row 3) normalised standard deviation, with the 20 individual sites (dots), and ideal agreement with observations (red line) and boxplot showing distribution. The UP corrections (selected for use in this study) have lower overall errors (cf. other methods) except SWdown, where no corrections to ERA5 are applied.

## 4 Data records

### 4.1 Data format

Timeseries data and site descriptive metadata are recorded in both plain text and netCDF4 (Rew et al., 1989) formats. Each site folder contains the following timeseries:

- [sitename]_raw_observations_[version]: site observed before project-wide quality control and gap filling (Table 1 gives period)





• [sitename]_clean_observations_[version]: after project-wide quality control and gap filling (Table 1 gives period)

- [sitename]_metforcing_[version]: continuous observations with reanalysis-derived data after quality control and gap filling (Table 2; forcing data for 10-year spin up, then Table 1 periods)

- [sitename]_era5_corrected_[version]: continuous timeseries (1990-2020) of bias corrected ERA5 reanalysis meteorological data (as used for gap filling and prepending metforcing observations)

Each site folder also contains the following site metadata:

- [sitename]_sitedata_[version].csv: comma separated text file for site characteristics metadata e.g. latitude, longitude, surface cover fraction, morphology etc. (Table 5, 6). This site characteristic data is also included within the metforcing netcdf (for convenience)

- index.html: A summary page of site information in html format, including site characteristics, site images, timeseries,
gap filling, quality control and diurnal plots

## 4.2 Timeseries metadata

The timeseries files include the following metadata:

- **title**: short description of the file

- **summary**: longer description of the file

• **sitename**: site code (e.g. AU-Preston)

- **long_sitename**: site long name, including city and country information

- **version**: version of current file

- **time_coverage_start**: start of timeseries in UTC (includes spin-up)

- **time_coverage_end**: end of timeseries in UTC

• **time_analysis_start**: start of observed (focus) period in UTC

- **time_shown_in**: time standard (always UTC)

- **local_utc_offset_hours**: offset in hours of local time from UTC

- **timestep_interval_seconds**: period of block averaging in seconds (timestep)

- **timestep_number_spinup**: number of timesteps prior to observed focus period

• **timestep_number_analysis**: number of timesteps in observed focus period

- **project_contact**: contact details for the Urban-PLUMBER project coordinators

- **observations_contact**: contact details of the observational site data providers

- **observations_reference**: published references associated with the observations

- **date_created**: date and time of creation of this file



• **source**: repository for processing code

• **comment**: additional comments associated with this dataset (e.g. excluded wind sectors).

The netCDF files also include the following metadata for each variable

• **long_name**: plain language description of variable

• **standard_name**: equivalent variable name under the CF (climate and forecast) conventions

• **units**: SI (international system) units

• **ancillary_variables**: name of the associated quality control flag variable

NetCDF files also include site characteristics parameter values and descriptions (Table 5). Note, the local_utc_offset does not account for day light savings. As the onset/finishing of day light savings causes distinct behaviour changes it is critical users of the data consider this in their applications.

**4.3 Site characteristics metadata**

Site characteristics (Table 5, 6) are essential for any use of these data, and fundamental to application of land surface models. These metadata are provided in two machine readable forms (plain text in csv files, and netCDF4). The metadata are primarily drawn from published sources, or as advised by the data providers. If local parameters are not known values are estimated from high resolution global datasets or derived from empirical relations. The sources for each parameter are included within 400 the site characteristic metadata.

There are numerous methods to estimate the probable extent and weighting of turbulent fluxes footprints relative to the eddy covariance sensors located on a flux tower (Velasco and Roth, 2010). The eddy covariance flux footprint provides a basis to identify which area (and weighting) should be used to estimate the land surface fractions impacting the measurements. Some studies in this collection determine the footprint and resulting land cover fractions dynamically (e.g. for each 30-min period), 405 whereas others assumed a constant radius (e.g. based on a climatological analysis or rule of thumb) (Table 6). Standardising the method to determine land cover fractions across sites is beyond the scope of this work, so users of metadata should be mindful of these differences.

Different methods for estimating surface roughness length and zero-plane displacement height can give significantly different values (Kent et al., 2017). Given different sites have derived values using different methods we also provide values using two 410 consistent morphometric methods (Macdonald (Macdonald et al., 1998) and Kanda (Kanda et al., 2013); Table 5, parameters 26 – 29) derived from surface fraction and building height parameters within the measurement footprint (Table 6). The Kanda modification to the Macdonald method accounts for the variability in roughness element height, resulting in larger displacement heights which are closer to estimates made with anemometric methods (Kent et al., 2017).The Macdonald method assumes that all the buildings have the same average roughness element height. However, care must be taken when using





Kanda values as some urban land surface models expect displacement height to be always lower than average building height (Hertwig et al., 2020).

Where not known, building height standard deviation ($\sigma_H$) is estimated from an empirical relation to building mean height ($H_{ave}$) (Kanda et al., 2013):

$$\sigma_H = 1.05 H_{ave} - 3.7. \tag{5}$$

Similarly, unknown local wall to plan area ratios ($\lambda_w$) are derived from roof area fraction ($\lambda_p$) and canyon height to width ratio ($H/W$) assuming an infinite canyon geometry (Masson et al., 2020):

$$\lambda_w = 2(1 - \lambda_p) H/W. \tag{6}$$

Frontal area index ($\lambda_f$) is sometimes reported in site literature without $H/W$ or $\lambda_w$, in which case these are estimated (again assuming an infinite canyon geometry) with (Porson et al., 2010):

$$\lambda_f = \frac{2}{\pi}(1 - \lambda_p) H/W. \tag{7}$$

Where not known or provided, mean annual anthropogenic heat flux (Varquez et al., 2021) or soil characteristics (Hengl, 2018a, b, c) are estimated from global datasets at 1 km or lower resolutions.

**Table 5:** Site characteristic metadata description and units. Select values are provided in Table 6.

| ID | Parameter | Units | Description |
|---|---|---|---|
| 1 | latitude | degrees_north | Latitude of tower |
| 2 | longitude | degrees_east | Longitude of tower |
| 3 | ground_height | m | Height above sea level of base of tower |
| 4 | measurement_height_above_ground | m | Height above ground level (agl) of eddy covariance equipment on tower |
| 5 | impervious_area_fraction | 1 | Plan area fraction of all impervious (hard) surfaces, including roofs, roads, paths and paved areas |
| 6 | tree_area_fraction | 1 | Plan area fraction of tree canopy (> 2 m) |
| 7 | grass_area_fraction | 1 | Plan area fraction of grass or other vegetation (< 2 m) |
| 8 | bare_soil_area_fraction | 1 | Plan area fraction of bare soil |
| 9 | water_area_fraction | 1 | Plan area fraction of water |
| 10 | roof_area_fraction | 1 | Plan area fraction of roofs ($\lambda_p$) |
| 11 | road_area_fraction | 1 | Plan area fraction of roads |
| 12 | other_paved_area_fraction | 1 | Plan area fraction of hard surfaces on ground excluding roads (e.g. paths, plazas, carparks etc) |
| 13 | building_mean_height | m | Mean height above ground of buildings ($H_{ave}$) |
| 14 | tree_mean_height | m | Mean height above ground of trees |
| 15 | roughness_length_momentum | m | Aerodynamic roughness length for momentum as reported in literature or provided by data providers |
| 16 | displacement_height | m | Zero-plane displacement height as reported in literature or advised by data providers |
| 17 | canyon_height_width_ratio | 1 | Mean building height to mean street canyon width (distance between buildings) ratio ($H/W$) |





| ID | Parameter | Units | Description |
|---|---|---|---|
| 18 | wall_to_plan_area_ratio | 1 | Sum of wall surface area to plan area ratio ($\lambda_w$) |
| 19 | average_albedo_at_midday | 1 | Median site albedo at midday (local standard time) for available observations |
| 20 | resident_population_density | person km$^{-2}$ | Resident (night) population density |
| 21 | anthropogenic_heat_flux_mean | W m$^{-2}$ | Anthropogenic heat flux annual mean |
| 22 | topsoil_clay_fraction | 1 | Clay fraction of topsoil |
| 23 | topsoil_sand_fraction | 1 | Sand fraction of topsoil |
| 24 | topsoil_bulk_density | kg m$^{-3}$ | Bulk (dry) density of topsoil |
| 25 | building_height_standard_deviation | m | standard deviation of building heights ($\sigma_H$) |
| 26 | roughness_length_momentum_mac | m | Aerodynamic roughness length for momentum calculated by the Macdonald morphometric method |
| 27 | displacement_height_mac | m | Zero-plane displacement height calculated by the Macdonald morphometric method |
| 28 | roughness_length_momentum_kanda | m | Aerodynamic roughness length for momentum calculated by the Kanda morphometric method |
| 29 | displacement_height_kanda | m | Zero-plane displacement height calculated by the Kanda morphometric method |

**Table 6:** Select site characteristic values (see Table 5 for definitions). Other site characteristic values and sources are provided within the collection (Lipson et al., 2022). Areas analysed for land cover fractions and roughness parameters are based on either a static radius around the flux tower (value given) or a dynamic footprint model (fpm), depending on the available site literature.

| Parameter | surface cover fraction assumption | ground_height | measurement_height_above_ground | impervious_area_fraction | tree_area_fraction | grass_area_fraction | bare_soil_area_fraction | water_area_fraction | roof_area_fraction | road_area_fraction | other_paved_area_fraction | building_mean_height |
|---|---|---|---|---|---|---|---|---|---|---|---|---|
| AU-Preston | 500m | 93 | 40 | 0.620 | 0.225 | 0.150 | 0.005 | 0 | 0.445 | 0.130 | 0.045 | 6.4 |
| AU-SurreyHills | 500m | 97 | 38 | 0.54 | 0.29 | 0.15 | 0.01 | 0.01 | 0.39 | 0.09 | 0.06 | 7.2 |
| CA-Sunset | fpm | 78 | 24.8 | 0.68 | 0.12 | 0.20 | 0 | 0 | 0.23 | 0.20 | 0.25 | 4.9 |
| FI-Kumpula | 1000m | 29 | 31 | 0.46 | 0.30 | 0.24 | 0 | 0 | 0.14 | 0.32 | 0 | 12.6 |
| FI-Torni | 1000m | 15.2 | 60 | 0.77 | 0.15 | 0.07 | 0 | 0.01 | 0.37 | 0.25 | 0.15 | 17.9 |
| FR-Capitole | 500m | 143 | 48.05 | 0.90 | 0.08 | 0.02 | 0 | 0 | 0.62 | 0.28 | 0 | 15 |
| GR-HECKOR | fpm | 30 | 27 | 0.916 | 0.040 | 0.016 | 0.010 | 0.019 | 0.516 | 0.201 | 0.199 | 11.3 |
| JP-Yoyogi | 500m | 39 | 52 | 0.92 | 0.06 | 0.01 | 0.01 | 0 | 0.41 | 0.32 | 0.19 | 9.0 |
| KR-Jungnang | 500m | 22 | 41.5 | 0.965 | 0 | 0.019 | 0.016 | 0 | 0.588 | 0.377 | 0 | 8.648 |
| KR-Ochang | 500m | 60 | 19 | 0.470 | 0.184 | 0.333 | 0.013 | 0 | 0.133 | 0.337 | 0 | 7.384 |
| MX-Escandon | fpm | 2240 | 37 | 0.94 | 0.06 | 0 | 0 | 0 | 0.57 | 0.37 | 0 | 9.69 |
| NL-Amsterdam | 500m | 0 | 40 | 0.68 | 0.15 | 0 | 0 | 0.17 | 0.44 | 0.07 | 0.17 | 14.2 |
| PL-Lipowa | fpm | 204 | 37 | 0.76 | 0.16 | 0.08 | 0 | 0 | 0.35 | 0.21 | 0.20 | 10.2 |
| PL-Narutowicza | 500m | 221 | 42 | 0.65 | 0.22 | 0.09 | 0.04 | 0 | 0.29 | 0.19 | 0.17 | 16 |





| | | | | | | | | | | | | |
|---|---|---|---|---|---|---|---|---|---|---|---|---|
| SG-TelokKurau06 | 1000m | 5 | 20.7 | 0.85 | 0.11 | 0.04 | 0 | 0 | 0.39 | 0.12 | 0.34 | 9.9 |
| UK-KingsCollege | fpm | 14.5 | 50.3 | 0.79 | 0.03 | 0.04 | 0 | 0.14 | 0.40 | 0.39 | 0 | 21.3 |
| UK-Swindon | 500m | 108 | 12.5 | 0.49 | 0.09 | 0.36 | 0.06 | 0 | 0.16 | 0.15 | 0.18 | 4.5 |
| US-Baltimore | 1000m | 157 | 37.2 | 0.313 | 0.536 | 0.138 | 0.007 | 0.006 | 0.160 | 0.153 | 0 | 5.6 |
| US-Minneapolis1 | fpm | 301 | 40 | 0.21 | 0.38 | 0.36 | 0 | 0.05 | 0.12 | 0.05 | 0.04 | 5.05 |
| US-Minneapolis2 | fpm | 301 | 40 | 0.05 | 0.2 | 0.73 | 0 | 0.02 | 0.01 | 0 | 0.04 | 5.05 |
| US-WestPhoenix | fpm | 340 | 22.1 | 0.48 | 0.05 | 0.10 | 0.37 | 0 | 0.26 | 0.22 | 0 | 4.5 |

## 4.4 Data flags

Each variable for each timestep has a quality control (qc) flag. For example, LWdown_qc lists qc flags for LWdown at each timestep. Flag numbers are consistent across all variables:

0.  observed by measurement at site and passes project quality control tests
1.  filled by observation: interpolated from site observations over short (2 h) periods OR filled by observations from nearby (< 10 km) stations over longer periods
2.  filled by ERA5: derived from ERA5 with site specific bias correction
3.  missing or removed through quality control (occurs only in timeseries without gap filling)

## 4.5 Wind sector exclusions

Turbulent flux data are excluded from certain wind directions (Table 7) because of:
- interference on flow from tower structure (as identified by data providers)
- markedly different land cover characteristics from sectors of interest (with guidance from data providers)

The US-Minneapolis site has different surface cover by wind direction but is retained in the collection because of its both a long observation period and its distinct land cover characteristics. Following previous studies (Menzer and McFadden, 2017) we subdivide this data into low-density residential area (northern sectors, US-Minneapolis2) and irrigated grassland with few built structures (south, US-Minneapolis1). Each are given their own site timeseries and metadata, resulting in 21 datasets.

**Table 7**: Site wind sector exclusions. Sites with sensible and latent heat fluxes excluded because of land cover or land use differences by wind sectors as described in the reference provided. Maps of these sectors are provided in the site data collection(Lipson et al., 2022).

| Sitename | Sectors excluded | Reason | Reference |
|---|---|---|---|
| FI-Kumpula | 0-180°, 320-360° | surface inhomogeneity | (Karsisto et al., 2016) |
| FI-Torni | 40-150° | flow interference from tower | (Järvi et al., 2018) |
| JP-Yoyogi | 170-260° | surface inhomogeneity | (Ishidoya et al., 2020) |
| US-Minneapolis1 | 75-285° | surface inhomogeneity | (Menzer and McFadden, 2017) |
| US-Minneapolis2 | 0-180°, 270-360° | 120-180°: flow interference from tower<br>270-360°, 0-120°: surface inhomogeneity | (Menzer and McFadden, 2017) |



## 5 Data availability

All site observational and characteristics data are openly available from https://doi.org/10.5281/zenodo.6590886 (Lipson et al., 2022) under a Creative Commons Attribution licence (CC-BY-4.0).

We recommend data users consult with site contributing authors and/or the coordination team early (i.e. planning stage) in projects that plan to use these data. Relevant contacts are included in site metadata.

## 6 Code Availability

Code used to process datasets are available at https://doi.org/10.5281/zenodo.6590942 (Lipson, M., 2022a).

Code used to create manuscript figures are available at https://doi.org/10.5281/zenodo.6590941 (Lipson, M., 2022b).

**7 Acknowledgements**

We would like to thank the vast number of people involved in day-to-day running of these sites that have been involved in instrument and tower installations (permitting, purchasing and site installation), routine (and unexpected event) maintenance, data collection, routine data processing and final data processing. We thank all those that have provided sites for the towers to be located and sometimes power and internet access. We acknowledge the essential funding for the instrumentation and other

infrastructure, for staff (administrative, technical and scientific) and students for these activities. We also thank those who offered data for use in this project which are not included at this time.

The project coordinating team are supported by UNSW Sydney and the Australian Research Council (ARC) Centre of Excellence for Climate System Science (grant CE110001028), University of Reading, the Met Office and ERC urbisphere 855005. Computation support from the ARC Centre of Excellence for Climate Extremes (grant CE170100023) and National

Computational Infrastructure (NCI) Australia. Contains modified Copernicus Climate Change Service Information. Site-affiliated acknowledgments are listed in Table 10.

**Table 8:** Funding acknowledgements for individual sites.

| Site | Contributing author | Site funding acknowledgements |
|---|---|---|
| AU-Preston | Andrew Coutts, Nigel Tapper | - |
| AU-SurreyHills | Andrew Coutts, Nigel Tapper | - |
| CA-Sunset | Andreas Christen, Oliver Michels | Canadian Foundation for Climate and Atmospheric Sciences (CFCAS, Project "Environmental Prediction in Canadian Cities (EPiCC)") and the Natural and Engineering Research Council of Canada (NSERC, RGPIN-03958, RGPAS-507854). Some instruments were supported by the Canada Foundation for Innovation (CFI, IF 2015, grant no. 33600) and BCKDF. We acknowledge the support of BC Hydro to operate the tower. |
| FI-Kumpula | Leena Järvi | ICOS Finland |



| FI-Torni | Leena Järvi | ICOS Finland |
|---|---|---|
| FR-Capitole | Valéry Masson | Météo-France and CNRS |
| GR-HECKOR | Nektarios Chrysoulakis | EU Horizon 2020 Research and Innovation Programme, under Grant Agreement No 870337 project CURE (http://cure-copernicus.eu) |
| JP-Yoyogi | Hirofumi Sugawara | Japan Society for the Promotion of Science KAKENHI grants (nos. 24241008, 15H02814, 18K01129, and 19H01975), and the Environment Research and Technology Development Fund (JPMEERF20191009) of the Environmental Restoration and Conservation Agency of Japan |
| KR-Jungnang | Jinkyu Hong, Sungsoo Jo, Yeon-Hee | Korea Meteorological Administration Research and Development Program "Development of Production Techniques on User-Customized Weather information" under Grant (KMA2018-00622) and National Research Foundation of Korea (NRF) grants funded by the Korean government (NRF-2018R1A5A1024958) |
| KR-Ochang | Jinkyu Hong, Je-Woo Hong, Keunmin Lee | Korea Meteorological Administration Research and Development Program under Grant KMI2021-01610 |
| MX-Escandon | Erik Velasco | National Institute of Ecology and Climate Change (INECC) and the Mexico City's Secretariat for the (SEDEMA) through the Molina Center for Energy and the Environment (MCE2). |
| NL-Amsterdam | Bert Heusinkveld | Netherlands Organisation for Scientific Research (NWO) Project 864.14.007 and the Amsterdam Institute for Advanced Metropolitan Solutions (AMS) project VIR16002. |
| PL-Lipowa | Wlodzimierz Pawlak, Krzysztof Fortuniak | - |
| PL-Narutowicza | Wlodzimierz Pawlak, Krzysztof Fortuniak | - |
| SG-TelokKurau06 | Matthias Roth | National University of Singapore (research grant R-109-000-091-112). |
| UK-KingsCollege | Simone Kotthaus, Sue Grimmond | EUfp7 Grant agreement no. 211345 (BRIDGE) and NERC ClearfLo (NE/H003231/1), NERC ARSF (GB08/19), EPSRC (EP/I00159X/1, EP/I00159X/2) and KCL. |
| UK-Swindon | Helen Ward, Jonathan Evans, Sue Grimmond | NERC  NE/H52479X/1 |
| US-Baltimore | Sue Grimmond, Ben Crawford | National Science Foundation (BCS-0095284, DEB-9714835) and USDA Forest Service. |
| US-Minneapolis | Joeseph McFadden | NASA Earth Science Division (NNG04GN80G) |
| US-WestPhoenix | Stevan Earl, Winston Chow | National Science Foundation (DEB-1832016) Central Arizona-Phoenix Long-Term Ecological Research Program (CAP LTER) |

## 7 Author Contributions

M.L., S.G. and M.B. conceived and coordinated the project, and prepared the protocols for contributing authors. M.L. collated
site datasets, wrote processing code, developed and undertook analysis of bias correction methods, prepared figures, prepared datasets and drafted the manuscript with guidance from S.G. and M.B. All other authors (listed alphabetically) collected primary data, prepared site information, processed datasets for inclusion in the collection and contributed to the manuscript. Table 8 lists contributing author site affiliation.

The authors declare that they have no conflict of interest.



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
