# Peer review of "Harmonized gap-filled datasets from 20 urban flux tower sites"

_Earth System Science Data, 2022_

## Author Comment (AC1)

**RC1: 'Comment on essd-2022-65', Anonymous Referee #1, 30 Jul 2022**

Lipson et al. present a unique harmonized dataset of 20 urban flux towers. The manuscript is clearly written overall, and the data files are adequately documented. I agree with the authors that the dataset is timely, critical, and can be of interest to many research and applications. However, I have a few minor comments that I suggest the authors consider.

Thank you for your review and helpful suggestions. Our responses follow.

1. Most parts of the manuscript focus on quality control, bias correction, and gap-filling the meteorological variables. It was unclear what was done specifically for flux variables (e.g., latent, sensible, momentum) until I opened up the files and examined the data. I think flux variables are quality-controlled and filtered but not filled. First, I suggest the authors make it clear in the main text. Second, I wonder why flux variables are not gap-filled. With net radiation, wind velocity, and roughness data available, it should be feasible to fill the flux variables. And, it could potentially facilitate the use of the dataset, esp. those who need gap-free data and those who need to aggregate to a coarse scale (e.g., daily, monthly).

We agree it is feasible to gap fill the flux variables with the data included in this collection. However, we note that the original motivation for producing this collection is for a model evaluation project in which it is undesirable to fill flux variables (as these variables are used to evaluate the performance of models). There are many methods that can be used to fill flux data, but an appropriate method will depend on research requirements. As the Urban-PLUMBER project had no requirements for gap-filled turbulent fluxes, we have made the data open and allow scientists to apply their preferred method or develop new methods.

To make it clear which variables are gap filled and which are not, we have updated the abstract:

*"Variables needed as inputs for land surface models (incoming radiation, temperature, humidity, air pressure, wind and precipitation) are quality controlled, gap-filled and prepended with 10 years of reanalysis-derived local data, enabling an extended spin up to equilibrate models with local climate conditions."*

and:

*"Other variables (turbulent and upwelling radiation fluxes) are harmonized and quality controlled without gap filling."*

The methods section and Table 2 are updated:

- Section 2.2: *"Turbulent fluxes and upwelling radiation fluxes are not gap filled (Table 2)."*
- Added a new column to Table 2 indicating which variables are gap filled
- Table 2 caption: *"Analysis and additional data are not gap filled."*

2. The proposed approach for bias correction seems robust and more flexible in capturing the diurnal variation than the linear regression approach. But, I think it requires enough data (e.g., multiple years) to generate a robust 'representative' curve, i.e., to smooth out the fluctuations and fill periods when no data are available for a specific hour of a day. Therefore, I suggest the authors add a brief description and justification of the method.

We agree. We have added in Section 2.6.1:

*"A 60-day rolling window is selected to smooth-out individual weather events while still capturing seasonal variation. Repeating the 'representative year' three times prior to smoothing ensures bias corrections match at the start and end of the year. The resulting set of bias correction curves (Fig. 4a) have greater robustness when multiple years are available."*

3. Line 393-395: Please be specific. Do all or any of the sites adjust for daylight saving time? Please consider providing the information if it's site-specific.

No sites account for daylight savings. This section, which was confusing, has been replaced simply with: *"Times in all datasets are UTC".*

4. Table 5: Consider providing some information on the spatial extents of each parameter. Some are based on tower location; some are averaged over the target or larger area.

Added to Table 5 caption: *"Parameters are determined for the turbulent flux footprint extent (Table 6), except 1-4 which are applicable to the tower itself, and 19 which is a function of the radiometer field of view (Offerle et al., 2003) and differs from the turbulent flux footprint (Schmid et al., 1991)."*

5. Table 6: Please provide the approximate spatial extent (e.g., in meters) for those using 'fpm'.

We have included the following in Table 6: *"or a dynamic footprint model (fpm). For the latter, the spatial extents are the order of a few hundred metres but are dynamic varying for example with atmospheric stability and wind direction".*

And the following in Section 4.3: *"determine the footprint and resulting land cover fractions dynamically (e.g. for each 30-min period based on that period's observed atmospheric variables such as stability and wind direction)"*

6. Data files: Please add a brief description of the data file structures (e.g., number of header lines, missing values), especially for those time series text files.

We have included in Section 4.2: *"Text file timeseries include metadata headers indicated with a hash (#) at line beginnings. Columns are headed by variable names in ALMA format (Table 2). NetCDF files include identical data, with additional attributes for each variable:"*

**RC2: 'Comment on essd-2022-65', Anonymous Referee #2, 6 Sep 2022**

The paper describes a dataset of eddy covariance data collected at 20 urban sites spread across Australia, Eastern Asia, Europe, and North America. Quality control and gapfilling methods applied to the data were described, as well as extended versions of driving climatic drivers based on reanalysis data harmonized with site data. This dataset is a unique and important contribution to eddy covariance, micrometeorology, and urban climate research.

Thank you for your review and helpful suggestions. Our responses follow.

The argument of using the dataset for climate models is somewhat limited by the fact that climate models mostly cannot take full advantage of datasets like this yet. However, the data presented here being available helps reduce the barriers for models to better take advantage of these data. The bias correction specific to urban environments is an innovative contribution, helping understand the mismatch between in-situ and larger scale representations of these sites; in particular, the hourly corrections are important for

analyses of short duration and very localized phenomena, which are not always well represented in these types of datasets.

*We agree that this data cannot be used to drive a full climate model. We have clarified this in the introduction:*

*"These [forcing] data can be used to drive land surface models offline at a single grid point. Other variables (turbulent and upwelling radiation fluxes) can be used to evaluative models in simulating land-atmosphere energy exchanges or in observational synthesis studies."*

One of the main limitations related to this dataset is that it was not made available within any of the many research networks that could have offered it a home, but rather it was made available via a general platform–Zenodo. Most notably, FLUXNET offers a clear path to community created datasets, with one such dataset having been described in this very journal: see Delwiche et al. 2021, ESSD. The dataset published in this manner not only makes it harder to find, but also makes it inherently less compatible with/comparable to similar datasets, increasing the difficulty of combined analyses. The datasets being useful for models is a desirable characteristic, as is being easy to use in synthesis-type analyses.

*We agree that making the data available through networks such as FLUXNET would expand the reach of this collection. There are several reasons why the data were not integrated with a network such as FLUXNET at this time. These include:*

- *The collection was generated for a specific purpose (evaluating urban land surface models) and therefore had particular variable, metadata, structure and formatting requirements which do not necessarily fit with existing networks (such as FLUXNET)*
- *FLUXNET requires user registration which limits the open availability of data*
- *Data can be integrated into networks such as FLUXNET (by us or others) in future*

*As such, at this time we will retain the current access through Zenodo, but thank the reviewer for articulating the advantages of integrating with a community network in future.*

Although the methods presented in the paper for gapfilling the data using different bias-corrections is very interesting, the comparison to alternative methods is not exactly a fair one. The goals for the ERA5 reanalysis dataset and, to some extent, also the WFDE5, are not to be used as single-point data streams, so the usefulness of the comparison of the methods proposed to these directly should not be interpreted as the proposed method being an improvement over these. The comparison to the FLUXNET2015 method, on the other hand, is very appropriate and led to interesting results. There are clear improvements from the linear method used for FLUXNET2015, which are clearly shown in the description. However, there are also variables/metrics that seem to not do quite as well, e.g., for short-wave radiation and, to some degree, wind variables. Exploring why these variables haven't seen the same level of improvement would be very interesting indeed, perhaps leading to some form of a hybrid approach.

*We agree with these comments and have added a paragraph in Section 3 to explain these issues and better compare with the other approaches (particularly the SW radiation, and why we decided not to apply bias corrections there):*

*"The other three methods (WFDE5 (W5), linear debiasing (LN) and the Urban-PLUMBER corrections (UP)) apply bias corrections to ERA5 data, and so may be expected to reduce ERA5 errors. However, W5 does not reduce errors at these sites, most likely because the*

*observations used for W5 bias correction are at very different spatial scales to the flux tower footprints (2500 km² versus 1 km²) and include observations from non-urban locations. The LN and UP methods eliminate spatial mismatches by drawing on local site or nearby rain gauge observations. As such, they do reduce the mean bias error to near zero for most variables. Notably, the UP methods outperform LN methods in normalised standard deviation. As Vuichard and Papale (2015) noted, linear methods do not conserve the variability of observations, nor can they correct for diurnal phase-shifts of some variables observed in urban areas (Figure 4). Therefore, we consider the UP methods to be the most appropriate at these urban sites. However, we apply no bias corrections to SWdown because the hourly and daily corrections (i.e. UP methods applied to other variables) adversely impact the standard deviation errors, as does the LN method for SWdown."*

Finally, some form of summary of the data might have been of interest in a paper like this. For example, a plot showing the carbon or energy fluxes through time for all 20 sites, maybe in a summary form? Or maybe fluxes grouped by climate or latitude or land cover-type? Maybe simple scatter plots of all data (or groups) of fluxes against driving variables (temperature or precipitation). Any of these would help highlight the coverage and usefulness of the dataset, which is quite clearly significant for someone who is familiar with the data, but maybe not as much to someone with more limited exposure to the domain.

We have included Figures 1 and 2 to help characterise the climatology of sites (e.g. the scatter of site air temperature and precipitation compared with global cities). However, we note that ESSD policy states "Although examples of data outcomes may prove necessary to demonstrate data quality, extensive interpretations of data – i.e. detailed analysis as an author might report in a research article – remain outside the scope of this data journal". As such, here we have kept summary information to a minimum with the expectation later synthesis studies will interpret and produce more detailed analysis like those suggested.

A few more specific details come next.

In the QC method presented, step 2 (setting nocturnal shortwave radiation set to zero) might seem like a reasonable approach, clearly no such thing as negative radiation. However, this might also inadvertently introduce a bias to the corrections: the sensor has its accuracy, which might lead to negative radiation values at night, and artificially forcing them to zero will reduce the impact of the sensor's accuracy limitations to the bias corrections that come later. Perhaps it is indeed better to have this step, but this limitation should at least be clearly stated.

Thank you for raising this. We have added in Section 2.3:

*"Note that quality control steps which eliminate observations at particular times (e.g. at night or after rainfall) can introduce biases (Grimmond, 2006) … Therefore, we also provide the "raw" observations (prior to quality control discussed here) in the collection as they may be more appropriate for some types of analyses."*

Still in the QC method, now for step 4, would it be necessary to adjust the number of standard deviations and/or window size for each variable? Wind varies in a very different way than atmospheric pressure, for example.

We follow a similar spike detection method to that described in Schmid et al. (2000) and Vickers and Mahrt (1996) (although with different thresholds, as their averaging periods are different). We agree the limit and window size is somewhat arbitrary (as stated explicitly in

Vickers and Mahrt, 1996) and might be better adjusted for different variables as suggested by the reviewer. However, when we undertook visual checks of rejected periods we found the results reasonable, so retain a standard threshold value for simplicity. The open collection includes plots of all rejected periods for all sites and variables, along with the raw data before QC in case researchers wished to use other methods.

To better explain these issues we have added the following to Section 2.3:

*"In addition, the outlier check values (Step 4) are somewhat arbitrary (as noted in Vickers and Mahrt, 1997). Therefore, we also provide the "raw" observations (prior to quality control discussed here) in the collection as they may be more appropriate for some types of analyses."*

Schmid et al., 2000: https://doi.org/10.1016/S0168-1923(00)00140-4

Vickers and Mahrt, 1996: https://doi.org/10.1175/1520-0426(1997)014%3C0512:QCAFSP%3E2.0.CO;2

A little more detail could have been given as to how the values for Table 4 were obtained, and also how they are used. The information currently listed in the paper is somewhat limited.

The following is added to Table 4:

*"Surface cover information as specified in ERA5 differs from actual tower site characteristics (see Table 6), and so ERA5 data is corrected (Section 2.6)."*

We also make clearer in the text that Table 4 shows the mischaracterisation of sites by ERA5:

*"To account for the mischaracterisation of sites (Table 4) and other listed deficiencies in ERA5… "*

In the hourly and daily corrections, section 2.6.1, step 4 involves extending 1-year worth of data by duplicating that year before and after; wouldn't this approach artificially create "jumps" towards the beginning and end of the dataset? Wouldn't it be better to use the actual years before and after the current year whenever available?

The 60-day smoothing ensures there are no jumps between years, however, we appreciate this was not clear in the text. We have introduced the following to Section 2.6.1:

*"A 60-day rolling window is selected to smooth-out individual weather events while still capturing seasonal variation. Repeating the representative year three times prior to smoothing ensures bias corrections match at the start and end of the year. The resulting set of bias correction curves (Fig. 4a) have greater robustness when multiple years are available."*

The evaluation of the gapfilling presented in section 3 focuses on the differences to ERA5. However, as mentioned above, the differences to the other methods, in particular the one for FLUXNET2015, would be of wider interest.

Thank you, as detailed in response to your previous comment, we have expanded the evaluation section to discuss these issues.

The statement regarding daylight savings in section 4.2 ("does not account for day light savings") is somewhat confusing: does this mean it is always UTC or that the conversion

from local time to UTC didn't correct for it so time might be shifted when DST? I believe it is supposed to be the former, however rephrasing that statement might help clarify this point.

Thank you, the details on daylight savings were confusing and have been replaced simply with: "Times in all datasets are UTC".

Finally, it would have been useful to have a netCDF file with data for all sites for easy access. This would be just a shortcut, but a very handy one.

Thank you for this suggestion. We have created a new netCDF file that includes all site observed data (after QC) and included this in the collection as a separately downloadable file [UP_all_clean_observations_v1.nc]. The advantage is that without spin up information it is less than 50 MB for all sites.

We have updated the repository doi link and documentation to include this new file:

https://doi.org/10.5281/zenodo.7104984